# Cross-Modal Object Detection Based on a Knowledge Update

**DOI:** 10.3390/s22041338

**Published:** 2022-02-10

**Authors:** Yueqing Gao, Huachun Zhou, Lulu Chen, Yuting Shen, Ce Guo, Xinyu Zhang

**Affiliations:** 1School of Electronic and Information Engineering, Beijing Jiaotong University, Beijing 100044, China; 18111069@bjtu.edu.cn; 2Center for Future Multimedia and School of Computer Science and Engineering, University of Electronic Science and Technology of China, Chengdu 611731, China; nechlu@163.com (L.C.); 202052080637@std.uestc.edu.cn (X.Z.); 3National Space Science Center, Chinese Academy of Sciences, Beijing 100190, China; rebeccashenstudy@yeah.net; 4University of Chinese Academy of Sciences, Beijing 100039, China; 5The 54th Research Institute of China Electronics Technology Group Corporation, Shijiazhuang 050081, China; Q631319563@yeah.net

**Keywords:** multimodality, multimodal encoder, graph convolutional network, knowledge update

## Abstract

As an important field of computer vision, object detection has been studied extensively in recent years. However, existing object detection methods merely utilize the visual information of the image and fail to mine the high-level semantic information of the object, which leads to great limitations. To take full advantage of multi-source information, a knowledge update-based multimodal object recognition model is proposed in this paper. Specifically, our method initially uses Faster R-CNN to regionalize the image, then applies a transformer-based multimodal encoder to encode visual region features (region-based image features) and textual features (semantic relationships between words) corresponding to pictures. After that, a graph convolutional network (GCN) inference module is introduced to establish a relational network in which the points denote visual and textual region features, and the edges represent their relationships. In addition, based on an external knowledge base, our method further enhances the region-based relationship expression capability through a knowledge update module. In summary, the proposed algorithm not only learns the accurate relationship between objects in different regions of the image, but also benefits from the knowledge update through an external relational database. Experimental results verify the effectiveness of the proposed knowledge update module and the independent reasoning ability of our model.

## 1. Introduction

The target of object detection is to locate the object from the complex image (video) background, separate the background, classify the object, and find the object of interest. Object detection has been widely applied in many fields, such as face detection, automatic driving, defect detection in engineering, crop disease detection, medical image detection, etc.

With the advance of multimodal learning, multimodal object detection has gradually become a popular research field. Multimodal learning aims at processing and understanding the available information from multiple sensory modalities. In recent years, with the rapid development of deep learning (DL), multimodal learning has drawn increasing attention. In particular, as an important branch of multimodal learning, cross-modal learning makes full use of the strategies of inter-modal representation, translation, and alignment in the multimodal learning field. The similarity between cross-modal learning and multimodal fusion is that the data of both come from all modalities, but the difference is that the data of the former are only available [1] in a certain modal, while the data of the latter are used for all modalities. Cross-modal retrieval, also known as cross-media retrieval, is one of the important applications of cross-modal learning which utilizes the data from all the modalities during the training process, but it uses single-type data in the testing process. Cross-modal retrieval aims to mine the information interaction between two different modalities, and its fundamental application is to explore the relationship between samples of different modalities, that is, retrieving the similar samples in another modal by a certain sample based on its semantics. In recent years, cross-modal retrieval has gradually become a frontier and hotspot of academic research, as well as an important direction of the information retrieval field. Cross-modal graphic retrieval is the most common research direction of cross-modal retrieval. For cross-modal image and text retrieval, keyword-to-image retrieval is called a pseudo-“cross-modal” problem because its essence is a match between the query keyword and the annotations of the image. Cross-modal image and text retrieval is based on both visual data and natural language description, so that more attention is needed in learning the interaction of the two modalities. Its purpose is to retrieve images through text (image) query without using any auxiliary information (text).

Cross-modal graphic retrieval can be divided into two forms: searches for text by image and searches for image by text. A previous study [2] has proposed that the semantic relationship between image and text can be defined as eight categories, which include irrelevant relationships, complementary relationships, interdependent relationships, anchoring relationships, illustration relationships, contrast relationships, poor illustrations, and poor anchoring relationships. In view of the complex semantic interaction between image and text, traditional cross-modal retrieval mainly uses statistical analysis methods, such as canonical correlation analysis (CCA) [3] and cross-modal factor analysis (CFA) [4]. Multimodal object detection is a branch of cross-modal graphic retrieval. The core problems of multimodal object detection are: (1) how to portray the interaction between text information and image information by machine learning models; and (2) how to establish a reasonable mapping between text and image. The classification diagram of the current research status of cross-modal graph retrieval based on deep learning is shown in Figure 1.

In recent years, with the powerful expressive capability of knowledge graphs, the research on building appropriate knowledge graphs has received more and more attention. Specifically, the introduction of human knowledge even becomes one of the most popular research directions of artificial intelligence (AI). Knowledge representation and reasoning provide AI systems with knowledge that they can process, making them capable of solving complex tasks, similar to humans. Knowledge graphs, which represent human knowledge in a structured form, have attracted extensive attention in both academia and industry in recent years. A knowledge graph is a structured representation of facts, entities, relationships, and semantic descriptions. Entities can be real-world objects or abstract concepts. Relationships represent the correlations between different entities. Semantic descriptions of entities and relationships contain predefined types and attributes. A property graph is a widely used type of knowledge graph in which nodes and relationships have their own properties.

The contributions of this work include the following three points:(1)Multimodal encoder

Three encoders, i.e., image encoder, text encoder, and multimodal encoder, are applied to extract text features and image features. Compared with the single-modal encoder, the proposed multimodal encoder learns a common low-dimensional space to embed images and text so that the image–text matching object can dig out rich feature information. Therefore, our model better understands the semantic meaning of images and text, and better aligns image features with text features, making it easier to learn the information across different modalities.

(2)GCN inference module

Instead of relying too much on labels like the conventional image–object detection models, our method introduces a GCN [5] inference module to use source knowledge for inference before obtaining the detection result. In this way, our method effectively fuses the multi-source information and performs the judgmental behavior, similar to humans.

(3)Knowledge update

In order to address the inconsistency between human understanding behavior and current algorithm design, our model further introduces an external knowledge base that is based on facts. To enhance the result of reasoning, our method associates each image area with an instance in the knowledge base, which helps better describe the relationship between different objects.

## 2. Related Works

Object detection can be divided into traditional detection methods and deep detection methods according to whether the idea of deep learning is applied. The main ideas of traditional detection methods can be roughly classified into several categories, i.e., frame difference method, background subtraction method, optical flow method, etc. Compared with traditional object detection algorithms, the deep detection methods not only apply deep networks to extract higher-level representation features of objects, but also integrate other machine learning tasks such as feature selection and classification into one model. There are two main types of deep detection algorithms, namely, two-stage object detection algorithms and single-stage object detection algorithms. Concretely, a two-stage object detection model is performed in two steps: (1) generate possible regions (region proposal) and extract image features with convolutional neural network (CNN); and (2) put the obtained features into the classifier for classification and position correcting. In 2015, Ren et al. [6] proposed the Faster RCNN algorithm, which developed the full network structure of two stages. In 2016, Dai et al. [7] published the region-based FCN (R-FCN) algorithm, which used a fully convolutional network (FCN) to achieve computational sharing and thus greatly improved the training speed. Due to the network structure characteristics of the two-stage algorithm, there is a bottleneck in its speed. Therefore, some researchers began to change their ideas and completed feature extraction, classification, and regression in one step. In 2016, Redmon et al. [8] proposed the you only look once unified (YOLO) algorithm. YOLO combines object determination and recognition, which greatly improves the speed of the algorithm.

For cross-modal graphic retrieval based on deep learning, to ensure retrieval accuracy, the semantic gap caused by the heterogeneous features of the underlying data among the modalities should be addressed. In addition, with the need for fast retrieval, the improvement of cross-modal graphic retrieval efficiency is also a current research hotspot. Based on the accuracy and efficiency of different cross-modal graphic retrieval methods, current popular algorithms can be divided into two categories: real-valued representation learning and binary representation learning. Real-valued representation learning methods usually have high accuracy and need to pay more attention to the semantic matching between images and texts, aiming to learn a real-valued public representation space in which the universal representation of the data of different modalities is real-valued. Binary representation learning, also known as cross-modal hashing, is usually used to speed up cross-modal retrieval. It maps different modal data to a common Hamming space, but the binarization process of such methods usually leads to an accuracy decrease in retrieval [9].

The early image information extraction was based on the description template. Specifically, the description sentence was generated by matching the language template with the recognized the object, as well as its attributes and relationships in the image. The authors of [10] selected the method of constructing a grammar tree to generate description sentences, Ref. [11] used triples to generate description sentences, and [12] selected phrases and then combined them into description sentences for image understanding.

With the development of this research field, an image comprehension method based on the similarity of the image and the image description sentence retrieval was proposed, that is, the image and the corresponding description sentence were mapped to the same feature space, and the description sentence was generated by calculating the similarity between the image and the sentence feature. In [13], the image and its corresponding sentences to two different feature spaces were mapped, then kernel canonical correlation analysis (KCCA) was employed to project the features to the same feature space. Finally, this method selected the description sentence by calculating the feature similarity.

With the rapid development of deep learning, some novel information fusion models that study how to measure the semantic similarity between image and text have been proposed. In the early stages of the research, two different networks were used to learn pictures and texts: for image feature extraction modules, a CNN network (such as Vgg or Resnet) is applied to extract image features, and for text features, an RNN (recurrent neural network) or BERT (bidirectional encoder representation from transformers) is employed to extract text features. Finally, the image and text features are transformed into the same semantic space through a fully connected network, and the matching of two cross-modal objects is determined by their cosine similarity or the distance. The authors of [14] first used deep learning methods to solve the problem of image information extraction, and applied multimodal recurrent neural networks to generate image description sentences. Image understanding is essential to the conversion of visual information to semantic information. To this end, inspired by the neural network-based encoder/decoder method in the machine translation [15], emerging methods regarded multimodal information matching as encoding visual information and decoding semantic information. Such an encoder–decoder framework had become the mainstream framework for image understanding. Generally, people use CNN to extract image feature vectors, and they input the image feature vectors into long-short term memory (LSTM) [16] to generate image description sentences.

Currently, the mainstream image–text matching methods can be grouped into three categories according to the different semantic structure associations between modalities, i.e., image–text alignment methods, cross-modal reconstruction methods, and image–text joint embedding methods. The image–text alignment method generally infers the potential alignment between sentence fragments and image regions by learning the relationship between the features of different modals that describe the same instance, and thus it achieves image–text matching. Cross-modal reconstruction methods pay more attention to global information. Such methods usually use one type of modal information to reconstruct the corresponding modal while retaining the reconstruction information, enhancing cross-modal feature consistency and semantic discrimination capabilities. The image–text joint embedding methods generally combine global and local information as the embedding of semantic features, so they learn more discriminative features. Meanwhile, such methods generally learn the relevance of image and text through the joint training of cross-modal data and the embedding of semantic features.

Wu et al. [17] proposed self-attention embeddings (SAEM), which exploit the self-attention mechanism to mine segment relationships in images or texts, and aggregate segment information into visual and textual embeddings. Wang et al. [18] proposed a novel position attention network (PFAN) to study the relationship between vision and text. In PFAN, the authors strengthened the joint embedding capability for image and text by focusing on the location information of objects. Li et al. [19] used a technique that captures key parts of images and text, thus generating visual representations that are easy for humans to interpret. First, all objects in the image are retrieved through a bottom-up attention mechanism. Then, the graph convolutional network is used to perform convolutional inference operations on part of the visual area to obtain a graph representation. The text and image are then aligned into the same embedding space via the gates and memory mechanisms in the long-short term memory network. Finally, a vector representation that represents the entire image is derived as the output of the whole network.

## 3. Method

During the research of this work, it was found that when high-level semantic concepts appear in images, ordinary neural networks often cannot provide reasonable relationships between objects. However, when people use natural language to describe what they see in pictures, these descriptions not only include objects and their attributes, such as texture, color, size, etc., but also their interactive information, such as relative position, and other high-level semantic concepts. In the human understanding process, visual reasoning about objects and semantics is the most crucial part. Therefore, if the machine wants to complete a variety of graphic object detection tasks, the reasoning ability is indispensable. However, existing simple image object detection only focuses on shallow information such as shape and size to detect the category and size of the object, resulting in a lot of high-level semantic information being ignored. Therefore, there is still a large space for improvement in the inference system of the existing visual text matching system. The multimodal object detection based on a knowledge update summarizes the reasons for the above problems because the reasoning results of the existing models are relatively simple and cannot match the deep relationship; thus, the reasoning results produced may be contrary to human common sense.

With the successful application of a transformer attention mechanism in natural language processing (NLP) learning [20] and pre-training technology based on a masked language model (MLM) in bidirectional encoder representations from transformers (BERT) [21], the attention module is used in many aspects, such as the aggregation and alignment of the embedded words in the sentence. Inspired by this fact, a multimodal object detection model based on a knowledge update is developed, which is an object detection model for visual language tasks. Focusing on the basis of image–text matching, semantic relations are used to assist object detection in our method. Meanwhile, associating with the local features of the image, our model lets the local feature regions of the image fuse with high-level semantic information. When people use natural language to describe what they see in a picture, these descriptions usually include not only objects that are relatively easy to recognize, but also their relative positions and other high-level semantic concepts, such as eating, reading, playing, etc. In this process, people will reason and judge the content in the image. If the machine wants to detect objects like a human does, it must have the same reasoning ability. Our multimodal object detection model based on a knowledge update is able to reason and judge based on the image content and then synthesize a semantic analysis.

The model proposed in this paper is mainly composed of four parts. The first part is the object relationship detection part, whose main work is to detect the objects appearing in the image. The second part is the multimodal encoder, whose role is to fuse image features and text features to obtain higher-level semantic features. The third part is the relationship modification part, which uses a GCN reasoning module that collectively evaluates all relevant facts before reaching the answer. The last part is the gated recurrent unit (GRU).

The framework of the overall model is shown in Figure 2.

### 3.1. Object Detection Module

In the object detection module, this paper implements a bottom-up attention mechanism, and each image can be represented as:(1)$$V=\left\{v_{1},v_{2},\ldots v_{k}\right\},v_{i}\in R^{D}$$
where $R^{D}$ represents a picture area block, D is the number of dimensions, k is the number of image area blocks, and $v_{i}$ is an image feature, which is a representation of a specific image area. Each feature is coded for a corresponding object or an area block with obvious object features in the image. This paper uses the bottom-up attention mechanism and the object detection network Faster R-CNN, and it uses the residual network ResNet-101 as the backbone of the network. For each category, the threshold of intersection over union (IOU) is set to 0.8. Then, the confidence threshold is set to 0.5 and all image regions where the probability of any object detection result is greater than the threshold will be chosen. Finally, the first 24 image regions with the highest level of detection confidence scores are selected. For each selected region, this paper extracts features after averaging the pooling layer to obtain features with a dimension of 2048, and then a fully connected layer is used to reduce the dimensions of obtained feature vectors. The purpose of this section is to divide the picture into different feature areas to pave the way for the next experiment. Through this part, the image characteristics corresponding to different regions can be obtained.

### 3.2. Multimodal Encoder

The purpose of this part is to fuse the regional features and text features of the image as nodes in the GCN. A six-layer transformer module is used for the text encoder and the multimodal encoder. Specifically, the text encoder and the multimodal encoder are initialized with the first six layers of BERTs. The input of the text encoder is each word from the input sentence, and some special elements are used to eliminate the ambiguity of different input formats. The information of each element can be adaptively aggregated by all the other elements according to the compatibility of their content, location, category, etc. After encoding the text information as $\left\{w_{0},\;w_{1},\;\ldots ,\;w_{n}\right\}$, the encoding results of the two encoders are fed into the multimodal encoder, which fuses the image features and text features via cross-attention. The finally obtained encoding features of the three encoders are used as the node feature V of the GCN. Through ablation experiments, it was found that the fusion feature of the multimodal encoder is about 5 percent better than using only image features. The cross-modal feature fusion process is shown in Figure 3.

### 3.3. GCN Relationship Modification

The graph convolutional network GCN inference module intends to use knowledge from multiple aspects to perform inferences before the result is obtained. Instead of relying too much on labels, as most image object detection models do, using the fusion information from multi-sources can give the model a judgmental behavior similar to a human. This paper develops an object detection model based on a knowledge update, that is, through a GCN.

Considering a series of facts to reason about the connections between different objects, the core idea of the graph convolution is to use edge information to aggregate node information and generate new node representations. In this paper, features obtained by the image encoder, text encoder, and multimodal encoder are the node features. The node features include image area knowledge information, text representation knowledge information, and joint representation knowledge information, which can fully reflect the fundamental characteristics of facts. Then, our model introduces an existing knowledge base to refine the relationship representation between nodes so that a better node representation to improve accuracy is achieved. The method of choosing a set of knowledge information from the knowledge base for training can help deal with the challenges posed by synonyms and homographs, as well as resolve the problem of how to ignore the main object. The proposed method focuses on learning the correct relationship between objects in different regions of the image and then completing the knowledge update through the external relationship database.

This paper builds an inference model based on an external knowledge base and enhances the representation of the relationship by considering the semantic relevance between image regions. The entity object $v_{i}\;\in \;E$ is represented by the image area in the used figure. Here, E represents the candidate objects represented by the detected image area, which is a real object. Our model uses their node representations as the input of the GCN. The GCN combines different regional objects to form a complete representation in the continuous iterative process. In more detail, the goal of the GCN is to learn how to combine the representation of the nodes of the graph, that is, how to represent the entire image as a connected graph $Gr=\left(V,\;E\right)$, where V is the set of detected regions and E denotes the relevance between image regions, describing the semantic relationship and association relationship between image regions. In this part, the external knowledge base will be used to update knowledge and improve the reasoning capability of the GCN module, thereby constructing a more reasonable relationship representation.

In order to solve the difference between human understanding behavior and current algorithm design, a fact-based relational reasoning model is introduced, which influences the existing representations through the common library, thereby enhancing the reasoning results. Common factual knowledge bases include Web Child [22], DBPedia [23], and Concept Net [24]. This article uses DBPedia as the candidate knowledge base. Different from the classic image-text matching method, the matching in this paper is used to update the inference information by analyzing the information in the image and referencing the facts of the knowledge base. As each inferred object is mapped to a relationship described by the implicit text, how to correctly choose to support the facts is of great importance to us. In order to be able to select facts more accurately, this paper proposes a learning-based method that embeds pairs of image text and facts in the same space and ranks the descriptions according to their relevance.

Using the external knowledge base DBPedia, the knowledge unit is represented as a triple expression form D={A, C, B}, where A is a visual feature based on the image, B is an attribute or phrase, and C is the relationship between A and B, which is given by the knowledge base. The relationship in the knowledge base is a subset of 13 pairs of possible relationships, where C can be equal to category which can be ‘comparable’, ‘possessing something’, ‘being something’, ‘having a certain attribute’, ‘being able to perform certain operations’, ‘desire’, ‘being associated with’, ‘being located’, ‘it is a certain part’, ‘action’, ‘used to do’, or ‘to be created’. The process of triples is performed by the toolkit provided by NLTK to split paragraphs into sentences, summarize the main idea of the sentence, identify the part of speech of these words, and split the words and other tasks. NLP is used to help machines understand human language. Since human language and the machine’s language are completely different, machines need to analyze sentences in a certain way, i.e., analyzing sentence components, and we must build a structure that allows the machine to roughly understand the meaning of the sentence. To this end, a sentence tokenizer named Punkt Sentence Tokenizer is used in this article. This sentence tokenizer can perform unsupervised machine learning, and it can be trained on any text. This study used this tool to divide a whole sentence into substrings. For example, the tokenizer can be used to find the words, text, and punctuation in a sentence and classify each word into nouns, verbs, prepositions, noun phrases, verb phrases, etc. In this way, our model not only converts text description sentences into triples, which correspond to the regions in the image, but also links the knowledge unit with the object detection results through continuous training iterations. Since DBPedia’s external knowledge base can form 190,000 knowledge units, different object detection images will correspond to different knowledge units, which can basically complete unlimited updates of knowledge.

We used the following two formulas to convert node features in different regions into vectors:(2)$$H\left(v_{i}\right)=w_{1}\left(v_{i}\right)$$
(3)$$G\left(v_{j}\right)=w_{2}\left(v_{j}\right)$$

Each image area I is associated with D in a knowledge base, which helps to better describe the relationship between entities. If a fact connects two different areas in the graph, then the two entities are connected. The GCN will jointly evaluate all the facts, which makes the method proposed in this article different from classification-based technology, as well as give the model the ability to automatically think and update. Through the knowledge update, a certain knowledge unit in the external knowledge base matches the relationship between two regions in the image, and a picture can be analyzed to obtain the corresponding description sentences and can be presented in the form of subject, predicate, and object. This paper uses a similarity score-based method to retrieve a set of related facts for a given pair of text images. To this end, one needs to measure the similarity between the detected objects in the visual area and the objects in the fact database. The similarity is calculated by converting them into vectors to find the cosine similarity between them. As some words may have some differences with the facts, this article obtains the fact scores by averaging the similarity scores of words. Our method sorts the facts according to the similarity score and selects the top 50 knowledge units of each object relationship. Then, the number of simultaneous occurrences of A and B in the knowledge unit is f, and f/50 is set as the confidence parameter P for training. Then, $E=P*H(v_{i})*G(v_{j})$ determines the edge feature. The final output feature representation depends on: (1) learnable weights (parameters $w_{1}$ and $w_{2}$ can be learned by back-propagation); and (2) in the adjacency matrices, $v_{i}$ and $v_{j}$ describe the point features of the graph structure. Then, a fully connected graph $Gr=\left(V,\;E\right)$ is obtained. This means if two image regions have a strong semantic relationship and are highly related, then there will be an edge with a high similarity score connecting these two image regions. Therefore, the core problem of this model is how to obtain the appropriate point feature matrix and train the parameter matrix $w_{1}$ and $w_{2}$.

The GCN consists of hidden layers composed of L nonlinear functions, specifically:(4)$$H^{\left(l\right)}=f\left(H^{\left(l-1\right)},A\right)=\partial \left({\tilde{D}}^{-\frac{1}{2}}\tilde{A}\;{\tilde{D}}^{-\frac{1}{2}}H^{\left(l-1\right)}W^{\left(l-1\right)}\right)\;\;\;\;\forall l\in \{1,\ldots ,L\}$$
where the input to the GCN is the feature matrix $H_{\left(0\right)}$ and relationship matrix A. $\tilde{A}=A_{adj}+I$, where $A_{adj}$ represents the adequate adjoint matrix of A and I represents the identity matrix. $\tilde{D}$ is the diagonal node degree matrix of $\tilde{A}$. ${\tilde{D}}^{-\frac{1}{2}}\tilde{A}{\tilde{D}}^{-\frac{1}{2}}$ is the symmetric normalized Laplacian, which is designed to solve the problems of numerical instability and exploding or vanishing gradients. $W^{l}$ is the trainable weight matrix of the layer of the GCN, and ∂ (∙) is the nonlinear activation function.

In particular, our model adds the remaining connections to the original GCN to obtain relation-enhanced representations of image region nodes:(5)$$V^{*}=W_{r}\left(RVW_{g}\right)+V$$
(6)$$V^{*}=\left\{v_{1}{}^{*},v_{2}{}^{*},\ldots v_{k}{}^{*}\right\},v_{i}{}^{*}\in R^{D}$$
where $W_{r}$ is the weight matrix of the residual structure and R is the similarity matrix of shape k×k. $W_{g}$ is the weight matrix of the GCN layer, whose dimension is D×D, and K is the matrix of the detected relation confidence. This paper normalizes the affinity matrix R row-by-row, as usual. The enhanced features $V^{*}$ are obtained through the GCN inference module and input into the GRU for the knowledge update.

### 3.4. GRU (Gated Recurrent Unit)

After the previous steps, our method already obtains the image representation with the relationship. Here, all the image area expressions are further linked together to get the overall visual representation. Specifically, this article executes this process by putting the regional feature sequence into the GRU [25] in order. The GRU is a type of recurrent neural network, which is similar to an LSTM. The GRU was designed to alleviate the relatively large amount of calculation required of an LSTM. It retains the advantages of an LSTM but saves time and computing complexity and is simpler than an LSTM. During the whole reasoning process, the description of the whole scene will be continuously updated and added in the storage unit of the hidden state. If an image relationship detection system can capture information well, it should also be able to generate a more appropriate image description. It will be helpful to train a better model by comparing the generated image description with the real image description. Finally, our model completes the object detection model based on the knowledge update.

The knowledge update means that during the whole reasoning process, the description of the entire scene will be continuously updated and added in the hidden state storage unit $h_{i}$. At each inference step i, the update gate $z_{i}$ analyzes the current input region features $v_{i}$ and the entire scene description of the hidden layer $h_{i-1}$ of the previous step, and calculates the degree to determine which unit needs to be updated. The calculation formula of the update gate is as follows:(7)$$z_{i}=\sigma_{z}\left(W_{z}v_{i}{}^{*}+U_{z}h_{i-1}+b_{z}\right)$$
where $\sigma_{z}$ is the sigmoid activation function. $W_{z}$, $U_{z}$, and $b_{z}$ are weights and biases. The next state will add new information to the previous state with the formula:(8)$${\tilde{h}}_{i}=\sigma_{h}\left(W_{h}v_{i}^{*}+U_{z}\left(r_{i}\circ h_{i-1}\right)+b_{h}\right)$$
where $\sigma_{h}$ is the tanh activation function. $W_{h}$, $U_{z}$, and $b_{h}$ are weights and biases. ∘ is an element-wise multiplication, which means that the corresponding elements are multiplied together. $r_{i}$ is the reset gate that decides what to forget based on the reasoning between $v_{i}$ and $h_{i-1}$. The computation of $r_{i}$ is similar to the update gate:(9)$$r_{i}=\sigma_{r}\left(W_{r}v_{i}{}^{*}+U_{r}h_{i-1}+b_{r}\right)$$
where $\sigma_{r}$ is the sigmoid activation function. $W_{r}$, $U_{r}$, and $b_{r}$ are weights and biases. Then, the description of the entire scene at the current step is a linear interpolation computed by the update gate $z_{i}$ between the previous description and the new content:(10)$$h_{i}=\left(1-z_{i}\right)\circ h_{i-1}+z_{i}\circ {\tilde{h}}_{i}$$
where ∘ is an element-wise multiplication. This is how our model accomplishes the knowledge update.

## 4. Experimental Evaluations

### 4.1. Data Set

The MS-COCO [26] data set was constructed by the Microsoft team. With its high quality, variety, and large quantity, it has become one of the most commonly used data sets in the computer vision field. The following are the characteristics of MS-COCO:

1. Each image has a corresponding object segmentation image in which the segmentation is accurate and concise.

2. It provides the contextual relationship of detection objects, which helps the model to better understand images.

3. Numerous and various objects are contained in the images of MS-COCO. Based on such sufficient information, the models trained by this data set have better robustness.

4. The data collection is large, with more than 300,000 pictures. As we all know, the larger the data sets, the higher the probability that the trained model will be effective.

### 4.2. Experimental Details

In this paper, the word embedding size is set to 300, and the dimension of the joint embedding space is set to 2048. The model in this paper is trained for 60 epochs. Specifically, the learning rate in the first 30 epochs is set to 0.00002 and is reduced to 0.000002 afterwards. In the evaluation of the test set, the model that achieves the best performance on the validation set is chosen.

Our algorithm aligns the object area information of the picture with the corresponding words, uses the multimodal encoder to extract cross-modal information, and then the GCN reasoning module is applied to establish the relationship between the regions. After that, our method introduces the knowledge update method in the GRU module. With the ability of autonomous reasoning, the proposed algorithm achieves better multimodal object detection results.

### 4.3. Display of Results

This section analyzes the overall results on the MS-COCO data set. As this paper innovatively aligns image objects with words in sentences, our model detects the key information of corresponding words in images. The advantage of our method is that it could automatically reason and match the words corresponding to the picture objects in a human-like manner. First, the overall model was tested without using the knowledge update, and it achieved an accuracy of 77.4%, which is already a good result. After that, under the same conditions, the overall model was tested with the knowledge update module, and the accuracy increased by 2.1%, which fully demonstrates the effectiveness of the knowledge update module. Images outside the training set are selected for testing, which verified the robustness and accuracy of our model.

### 4.4. Effect Display of Adding the GCN Inference Module

#### 4.4.1. The Comparison Heat Map of the Generalized Category Detection of the Object

The result of object detection is not limited to accuracy, which can be also observed through the semantic relationship. Figure 4 and Figure 5 show the comparison heat maps of the results of the generalized category detection of the object.

#### 4.4.2. Detection of Behavioral Information

As the capability of object detection has been extended to the detection of semantic relations, the application of our method is no longer limited to classifications such as the single-modal object detection models. Concretely, even behavior detection, position detection, and higher-level semantic information detection can be achieved by our model. Detection of behavioral information is shown in Figure 6 [27].

#### 4.4.3. This Model Has the Function of Further Reasoning

Using external knowledge bases to update knowledge can make our GCN reason more accurately and extensively. The model itself can think about the relationship between the object and the properties of the object, the behavior of the object, etc. It is as though our model has a mind of its own, jumping out of the label’s limitations, and detecting more information, as shown in Figure 7 [28].

Figure 6 and Figure 7 are randomly selected test samples to prove the robustness and accuracy of the model.

## 5. Summary

In this paper, a novel cross-modal object detection method based on knowledge updating is proposed. Concretely, this model consists of four parts: the object detection module, the GCN relationship modification, the multimodal encoder, and the GRU. The highlight of our method is its ability to take full advantage of the connections between different areas of the image and update the knowledge with an external knowledge base. In our method, the attributes of objects can be detected well through the associated objects, which enables the model to analyze and judge independently. Experimental results have proven that the multimodal object detection model based on knowledge updating not only improves accuracy, but also has the ability to reason autonomously.

## Figures and Tables

**Figure 1 sensors-22-01338-f001:**
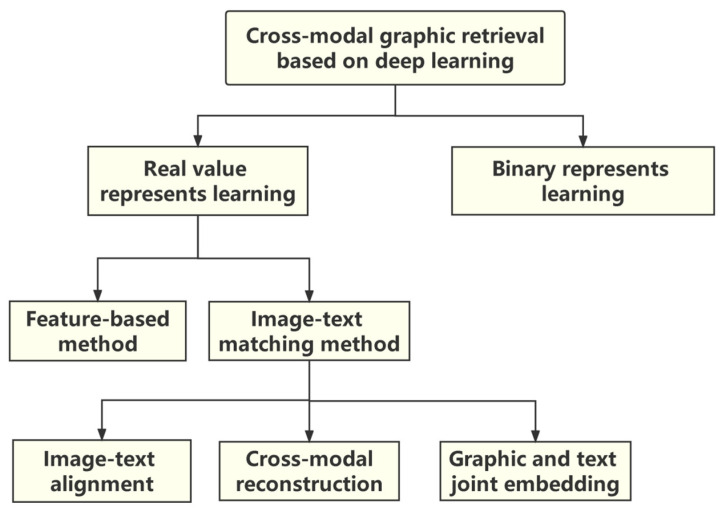
The classification diagram of the current research status of cross-modal graphic retrieval based on deep learning. Cross-modal graphic retrieval based on deep learning includes real valued-representation learning and binary representation learning. Real valued-representation learning is divided into two areas: the feature-based method and the image–text matching method. The image–text matching method includes image–text alignment, cross-modal reconstruction, and graphic and text joint embedding.

**Figure 2 sensors-22-01338-f002:**
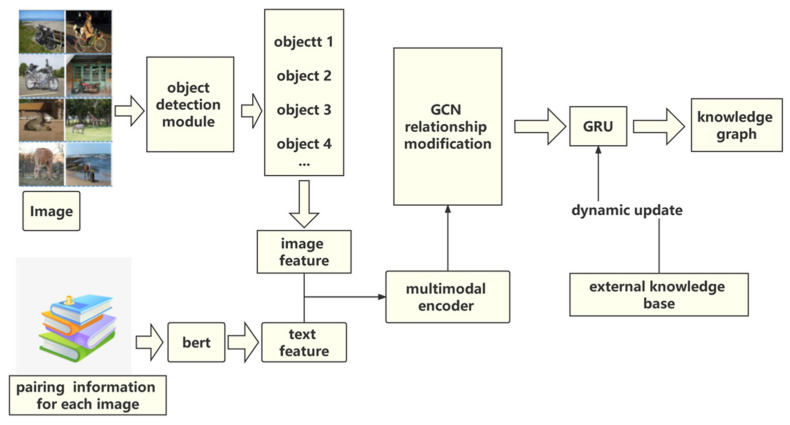
This model consists of four parts: the object detection module, the GCN relationship modification, the multimodal encoder, and the gated recurrent unit (GRU). The role of the object detection module is to extract different areas of the image. The purpose of the GCN relationship modification is to establish the relationship between different regions of the image. The multimodal encoder obtains node features more perfectly, and the GRU is an iterative, updated module.

**Figure 3 sensors-22-01338-f003:**
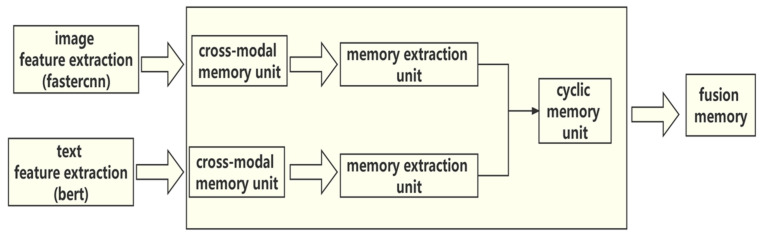
The step-by-step process of how image features and text features are fused, which is an important part of multimodal encoders.

**Figure 4 sensors-22-01338-f004:**
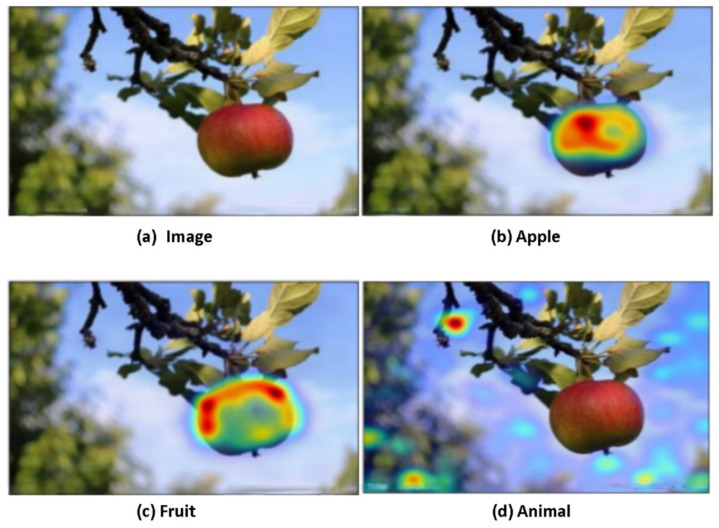
The proposed knowledge-updated multimodal object detection model can detect the types of objects. (**a**) is the original image used for object detection verification. (**b**) is the heat distribution map of the region of interest when the word “apple” is input to the model. (**c**) is the heat distribution map of the region of interest when the word “fruit” is input to the model. (**d**) is the heat distribution map of the region of interest when the word “animal” is input to the model. According to the above results, the model can clearly distinguish the category, name, and location of the object. This model can think and judge independently; an apple is a fruit, but not an animal.

**Figure 5 sensors-22-01338-f005:**
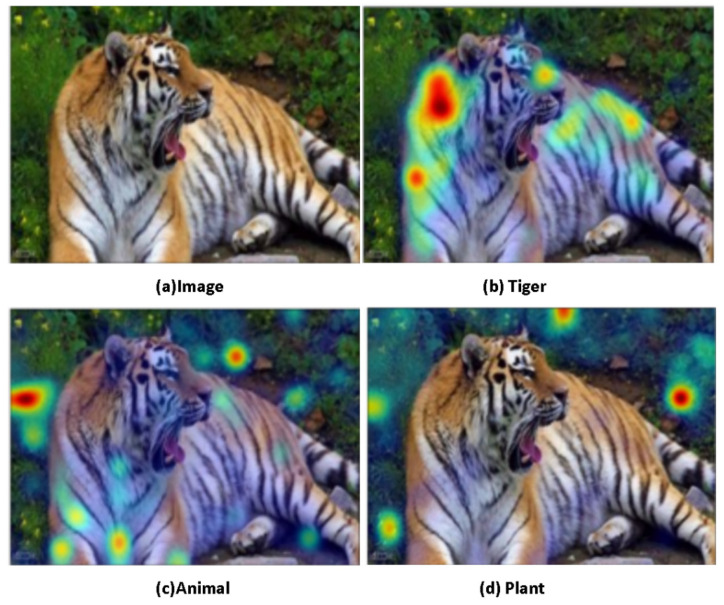
(**a**) is the original image of object detection verification, while (**b**–**d**) are the object detection heatmaps when the words “tiger”, “animal”, and “plant” are entered. (**b**) is the heat distribution map of the region of interest for the detection of the word “tiger”, (**c**) is the heat distribution map of the region of interest for the detection of the word “animal”, and (**d**) is the detection of the word “plant”. According to the verification results, this model already has the ability to analyze and reason. Not only can our model detect the tiger as the area of interest, but it can also recognize that the tiger is an animal, rather than a plant.

**Figure 6 sensors-22-01338-f006:**
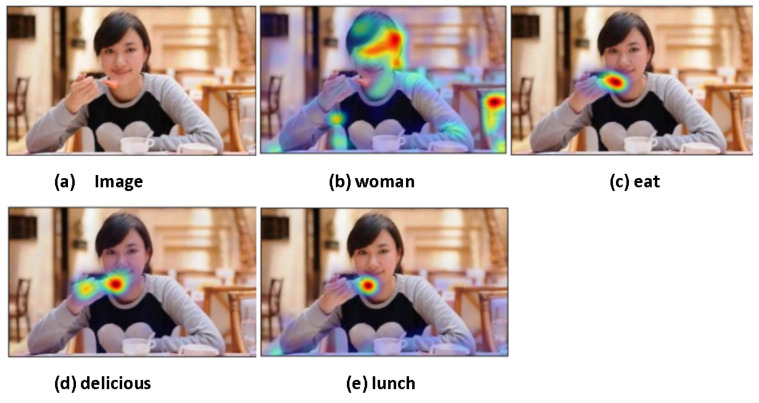
(**a**) is the original picture. The following sentence is input: woman is eating delicious lunch. Our model can be used to detect this sentence word by word, and the heat maps are shown in the figures above. Obviously, our model can find the region of interest corresponding to the text. Meanwhile, our model can not only detect the objects, but also detect their position, behavior, color, and other advanced attributes.

**Figure 7 sensors-22-01338-f007:**
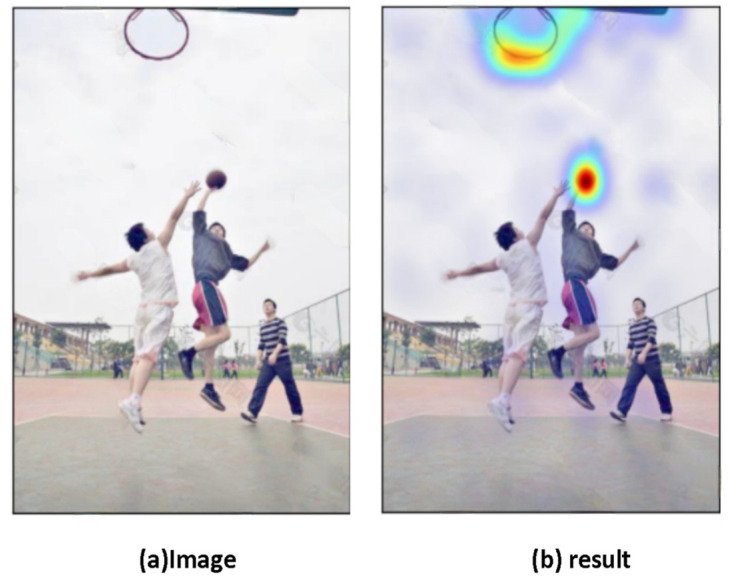
By using an external knowledge base to update knowledge, our model has the ability to reason and judge more broadly. (**a**) is the original picture. As the word basketball is input, our model will detect basketballs and basketball-related items, i.e., the basket. This proves that with the assistance of the knowledge base, our model has the ability to judge and classify independently.

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
