# Peer review of "Cross-Modal Object Detection Based on a Knowledge Update"

_sensors, 2022, doi:10.3390/s22041338_

Round 1

Reviewer 1 Report

An interesting cross-modal object detection method based on knowledge update is proposed in this paper. The contributions are clear but there are some problems needed to consider for the improvement of the current version.

  1. All the first appeared abbreviations are to be written out. Such as GCN in line 28, NLP and MLM in line 185, BERT in line 186 etc.
  2. The presentation in line 60 “the annotation annotations…” and line 298 “C is two entities A The relationship between B and B…” should be corrected.
  3. Some figures in this paper are recommended to be replaced by vector graphics.

4. The process of knowledge update should be described detailedly.

Author Response

Comments: An interesting cross-modal object detection method based on knowledge update is proposed in this paper. The contributions are clear but there are some problems needed to consider for the improvement of the current version.

Response: Thanks so much for your careful reading and encouraging comments.

Comments: 1. All the first appeared abbreviations are to be written out. Such as GCN in line 28, NLP and MLM in line 185, BERT in line 186 etc.

Response: Thank you for raising this concern. We have made changes based on your comments. We give full name for each abbreviation.

Comments: 2. The presentation in line 60 “the annotation annotations…” and line 298 “C is two entities A The relationship between B and B…” should be corrected.

Response: Thank you for your careful reading and comments. We have corrected these typos in the revision.

Comments: 3. Some figures in this paper are recommended to be replaced by vector graphics.

Response: Thanks for your suggestion. We've repainted parts of the image to improve clarity. We've repainted parts of the image to improve clarity. For the problem of image clarity, in order to prove the robustness of our experimental model, the test images we choose are downloaded from the Internet, and we show the clarity of the original image.

Comments: 4. The process of knowledge update should be described detailedly.

Response: Thank you for your comments. We have added a detailed reasoning description in the knowledge update section to the paper, please refer to Section 3.4 for more details.

Reviewer 2 Report

Dear Editor and Dear Authors,

The topic of this paper is very interesting and thank you for your trust to send me the manuscript to review it.

The comments and suggestions can be find below:

  1. The authors used personal pronouns in the paper, this is unusual for scientific papers. The authors should use Passive Voice instead, since this is mandatory for well written scientific papers in high quality journals.
  2. The Abstract must be purposeful and concise, currently the Abstract is not appropriate. Please rewrite it.
  3. Further, there abbreviations in the Abstract, this is unusual. Also, the “GCN” abbreviation is not explained at the first appearance. All the abbreviations should be explained in the manuscript at the first appearance.
  4. The figures are very blur and hazy, this is unacceptable. The equations are unacceptable too. All the figures and equations should be corrected/rewritten.
  5. The Introduction section is not concise, and it does not contains the main information related to the research and the manuscript itself.
  6. The Related Work should be expanded with more new and valuable references. Also, the reference list is not appropriate.
  7. The method is not presented appropriately. It is very hazy and unclear, specially for new readers who are not familiar with the topic, also the Experiments section too. The whole section should be clarified. The conclusion should be written more specified, aligned with the Abstract and with the manuscript content.
  8. There are missing parts required by MDPI in the end of the paper. Please, check the paper.

The paper has serious drawbacks that should be corrected. The paper should be revised in major, and submitted again.

Recommendation: Major revision

Author Response

Comments: 1:The authors used personal pronouns in the paper, this is unusual for scientific papers. The authors should use Passive Voice instead, since this is mandatory for well written scientific papers in high quality journals. The Abstract must be purposeful and concise, currently the Abstract is not appropriate. Please rewrite it.

Response: We have followed your suggestion to rewrite the Abstract section. We have streamlined the abstract, added a background description, and described the algorithm in more details.

Comments: 2:Further, there abbreviations in the Abstract, this is unusual. Also, the “GCN” abbreviation is not explained at the first appearance. All the abbreviations should be explained in the manuscript at the first appearance. The figures are very blur and hazy, this is unacceptable. The equations are unacceptable too. All the figures and equations should be corrected/rewritten.

Response: Thank you for comments. We've dealt with the issue of keywords not giving their full names. For the problem of image clarity, in order to prove the robustness of our model, the test images we choose are downloaded from the Internet, and we show the clarity of the original image. For the question of formulas, we have rewritten and numbered them using the formula editor.

Comments: 3:The Introduction section is not concise, and it does not contains the main information related to the research and the manuscript itself. The Related Work should be expanded with more new and valuable references. Also, the reference list is not appropriate.

Response: Following your suggestion, we have revised the introduction part. We added the introduction of object detection and several literatures at the beginning; then moved to multimodality. Because the existing multimodality cannot make good use of knowledge, effective use of external knowledge becomes a popular direction . We provide a brief overview of the newly added literature in the Related work.

Comments: 4:The method is not presented appropriately. It is very hazy and unclear, specially for new readers who are not familiar with the topic, also the Experiments section too. The whole section should be clarified. The conclusion should be written more specified, aligned with the Abstract and with the manuscript content.

Response: We have added descriptions and formulas, refer to Section 3. The conclusion and abstract have been improved correspondingly.

Comments: 5:There are missing parts required by MDPI in the end of the paper. Please, check the paper. The paper has serious drawbacks that should be corrected. The paper should be revised in major, and submitted again.

Response: Thanks for your careful reading. The paper lacks information, the author of the submitted paper needs to check it.

Reviewer 3 Report

The authors propose a multi-model object recognition model based on knowledge updates.

The text needs careful revision since there are many typos and incorrections.

Figure 1 is introduced in the Introduction section but only explained in section 2. You should explain it in section 1 or move the picture to section 2.

In general, the figures have low definition. Should be improved.

Section 2 about related work is incomplete. For example, the work you compare with [22] is not mentioned here. You should also explain the differences between related works and how you improve them. How do you compare, for example, with the work from [22]? It is important to understand why your work improves the state of the art.

The models, networks, etc. must be fully described and explained. If I want to replicate your solution, the given description is not enough. For example, the GCN and the GRU should be detailed.

The algorithm should be somehow illustrated during the explanation. This would allow the reader to better understand some details of the dataflow.

In the results section, you must explain the training process more carefully. The number of instances used to train each model, how did you proceed with the evaluation, etc. You should explain why is your solution better. What aspect has contributed mostly to the mentioned improvements? It is also important to include the computational complexity of the method. Did your solution improve the accuracy at the cost of some higher computational complexity?

Author Response

Comments: 1:The text needs careful revision since there are many typos and incorrections. Figure 1 is introduced in the Introduction section but only explained in section 2. You should explain it in section 1 or move the picture to section 2.In general, the figures have low definition. Should be improved.

Response 1 : We explained Figure 1 in Section 1, and added some descriptions. The eight terms in the figure are briefly introduced in the caption of Figure 1. The pictures have also been updated.

Comments: 2:Section 2 about related work is incomplete. For example, the work you compare with [22] is not mentioned here. You should also explain the differences between related works and how you improve them. How do you compare, for example, with the work from [22]? It is important to understand why your work improves the state of the art.

Response 2:We introduce [16] (ref.16 in the revision) in the related work section and briefly describe the differences between this paper and it and add several related papers.

Comments: 3:The models, networks, etc. must be fully described and explained. If I want to replicate your solution, the given description is not enough. For example, the GCN and the GRU should be detailed. The algorithm should be somehow illustrated during the explanation. This would allow the reader to better understand some details of the dataflow.

Response 3:We describe the details of the work in parts such as GCN and GRU. The details of the work are in the Section 3.3 and Section 3.4.

Comments: 4:In the results section, you must explain the training process more carefully. The number of instances used to train each model, how did you proceed with the evaluation, etc. You should explain why is your solution better. What aspect has contributed mostly to the mentioned improvements? It is also important to include the computational complexity of the method. Did your solution improve the accuracy at the cost of some higher computational complexity?

Response: 1) In Section 4.2, we describe in detail what should be done step by step based on the dataset.

2) How to calculate the improvement of accuracy. The most similar to our work is [22]; but it is doing sentence matching, and our work is doing word matching, and cannot be directly compared. So we apologize the previous accuracy comparison with [22] is not correct and has been deleted. To verify the effectiveness of the knowledge update, we compare the accuracy of our target detection before using the knowledge update with the accuracy of the object detection after using the knowledge update. After adding the knowledge update, there is indeed a significant improvement in the effect. The accuracy of our multimodal object detection model is already relatively high; after adding knowledge, it is further improved.

3) Complexity analysis: We use GCN, GRU, Fast RCNN modules in order to make the model have the ability to reason independently. So the complexity depends on the abovementioned modules. In addition, we use the word segmentation software NLTK, which is efficient. Although it has extra cost, it is unavoidable for doing the new task of this paper.

Round 2

Reviewer 2 Report

Dear Editor and Dear Authors,

The comments and suggestions can be find below:

  1. The manuscript is much better now, but still the diagrams are very blur and hazy, this is unacceptable! The images inside the figures are not denoted properly, i.e. (a), (b), etc. Also, the images are not of same size inside the figure!
  2. The equations are unacceptable too. They are not of same size, and they are not clear. In equation (2), the ½ is in the exponent, or not? Did the authors use an equations editor, or something similar, or the equations were written with simple text editing tools? These are serious mistakes, and they are unacceptable for serious scientific paper and for journal such as MDPI Sensors.
  3. All the figures and equations should be corrected/rewritten!
  4. The Method section is better now, but the unclear equations can cause ambiguity.
  5. The paper still contains many personal pronouns in the paper, the paper is not rewritten entirely. Also, still there language mistakes in paper. Maybe the authors should send the paper to language editing service if they cannot deal with the English.

The paper is now better, but still it has serious drawbacks (i.e. unclear equations and diagrams) that should be corrected. The paper should be revised, and submitted again.

Recommendation: Revision

Author Response

Comments 1:The manuscript is much better now, but still the diagrams are very blur and hazy, this is unacceptable! The images inside the figures are not denoted properly, i.e. (a), (b), etc. Also, the images are not of same size inside the figure!

The equations are unacceptable too. They are not of same size, and they are not clear. In equation (2), the ½ is in the exponent, or not? Did the authors use an equations editor, or something similar, or the equations were written with simple text editing tools? These are serious mistakes, and they are unacceptable for serious scientific paper and for journal such as MDPI Sensors. All the figures and equations should be corrected/rewritten!

Response: Thanks so much again for your comments. We have rewritten all the equations and redrawn all the figures. We think now the equations and figures satisfy the conditions of Journal. In the revision, we used (a), (b), etc. to distinguish the sub-images in the image. We've updated all equations with the equation editor, and the numbers are written to make sure they're clear, concise, and have the same size. In Equation (4) in the revision, 1/2 is indeed in the exponent, and we are sorry for the confusion for the reader. Hopefully, our updated equations will no longer confuse readers.

Comments 2: The Method section is better now, but the unclear equations can cause ambiguity. The paper still contains many personal pronouns in the paper, the paper is not rewritten entirely. Also, still there language mistakes in paper. Maybe the authors should send the paper to language editing service if they cannot deal with the English.

Response: Thanks a lot for raising these concerns and for your patience. We have made a great effort to improve the English writing. Hopefully it now satisfies the needs of a high-quality scientific paper. In addition, the personal pronouns have all been corrected.

Reviewer 3 Report

The manuscript has been improved according to the reviewer's comments. However, there are still some problems with figures and equations editing.

The paper needs detailed language editing. The authors should also avoid using personal pronouns.

Author Response

Comments 1:The manuscript has been improved according to the reviewer's comments. However, there are still some problems with figures and equations editing.

Response: Thanks a lot for your comments. We have rewritten all the equations and redrawn all the figures. We think now the equations and figures satisfy the conditions of Journal. In the revision, we used (a), (b), etc. to distinguish the sub-images in the image. We've updated all equations with the equation editor, and the numbers are written to make sure they're clear, concise, and have the same size.

Comments 2:The paper needs detailed language editing. The authors should also avoid using personal pronouns.

Response:Thank you very much for raising this concerns. We have made a great effort to improve the English writing. Hopefully it now satisfies the needs of a high-quality scientific paper. In addition, the personal pronouns have all been corrected.